

# Potential of global land water recycling to mitigate local temperature extremes

Mathias Hauser[1], Wim Thiery[1, 2], and Sonia Isabelle Seneviratne[1]

[1]Institute for Atmospheric and Climate Science, ETH Zurich, Zurich, Switzerland
[2]Vrije Universiteit Brussel, Department of Hydrology and Hydraulic Engineering, Pleinlaan 2, 1050 Brussels, Belgium

**Correspondence:** Mathias Hauser (mathias.hauser@env.ethz.ch)

**Abstract.** Soil moisture is projected to decrease in many regions in the 21st century, exacerbating local temperature extremes. Here, we assess the potential of keeping soil moisture conditions at historical levels in the 21st century by 'recycling' local water sources (runoff and a reservoir). To this end, we develop a 'land water recycling' (LWR) scheme which applies locally available water to the soil if soil moisture drops below a predefined threshold (a historical climatology), and assess its influence
on the hydrology and extreme temperature indices. We run ensemble simulations with the Community Earth System Model for the 21st century and show that our LWR scheme is able to drastically reduce the land area with decreasing soil moisture. Precipitation responds to the LWR with increases in mid-latitudes, but decreases in monsoon regions. While effects on global temperature are minimal, there are very substantial regional impacts on climate. Higher evapotranspiration and cloud cover in the simulations both contribute to a substantial decrease in hot temperature extremes. These reach up to about 1 °C regionally,
and are of similar magnitude as the regional climate changes induced by a 0.5 °C difference in the global mean temperature, e.g. at 1.5 °C vs. 2 °C global warming).

## 1 Introduction

Land water plays an important role for temperature extremes and heatwaves. Changes in soil moisture (SM) can alter the
amount of water that is available for evapotranspiration (ET), affecting local climate through its impact on the energy and water cycles (Seneviratne et al., 2010). The relationship between SM and temperature has been studied extensively in observation-based (Hirschi et al., 2011; Whan et al., 2015) as well as model (Fischer et al., 2007; Lorenz et al., 2010; Hauser et al., 2016) studies.

    Projections for the 21st century show decreasing SM in many mid-latitude regions (Orlowsky and Seneviratne, 2013; Berg
et al., 2017), although the trends can differ substantially between models (Lorenz et al., 2016). The effect of future SM trends on temperature is typically assessed with idealised sensitivity experiments where SM is prescribed to pre-defined values (Koster et al., 2004; Seneviratne et al., 2013; Hauser et al., 2017). In these climate model experiments, simulations with interactive



SM are compared to simulations where SM is held at historical levels. A multi-model assessment found substantially reduced temperature extremes in projections with historical SM levels (Seneviratne et al., 2013; Lorenz et al., 2016; Vogel et al., 2017). A separate single-model experiment by Douville et al. (2016) came to similar conclusions, identifying that in regions with decreasing SM, it is responsible for up to one third of the projected increase in temperature extremes during the 21st century.

However, these sensitivity experiments do not conserve water (Hauser et al., 2017), since moisture is artificially added or removed from the soil if it gets too dry or too wet.

More reality-grounded experiments on potential effects of the land water cycle on climate can be obtained with simulations assessing the influence of irrigation. Irrigation is a land management practice that applies water to the soil, elevating SM levels. Therefore, irrigation does not only help to sustain global food production, by providing agricultural crops the necessary water

to grow, but it also influences local weather and climate. There is a number of studies investigating the impact of irrigation on climate with global climate models (Sacks et al., 2009; Cook et al., 2011; Guimberteau et al., 2012; Krakauer et al., 2016; de Vrese et al., 2016; Hirsch et al., 2017; Thiery et al., 2017). For mean temperatures, most studies report a small cooling effect. However, temperatures often show an asymmetric response to irrigation. While local annual minimum temperatures (TNn) may even slightly increase, the annual maximum (TXx) shows a much larger response than the mean. TXx was found

to decrease by $-0.78\,°C$ averaged over all irrigated land area, and up to $-2\,°C$ regionally (Hirsch et al., 2017; Thiery et al., 2017). The asymetric effect of irrigation is because more water is applied during warm and dry periods, when the effect of SM on surface temperatures is especially pronounced (e.g. Schwingshackl et al., 2017). Therefore, irrigation has the potential to alleviate heat waves, and it has been proposed that its potential effects on local to regional scale should be better factored in within the context of mitigation and adaptation scenarios (Hirsch et al., 2017).

However, irrigation uses large quantities of water. The estimated water consumption for irrigation has risen from approximately $600\,km^3\,yr^{-1}$ in 1900 to more than $2000\,km^3\,yr^{-1}$ in the year 2000 (Döll and Siebert, 2002; Wisser et al., 2010). Indeed, irrigation is responsible for $70\,\%$ of the global freshwater use by humans, and a large fraction of the irrigation is realised with groundwater (Siebert et al., 2010; Döll et al., 2012). Overuse of this water resource can lead to groundwater depletion in intensely irrigated regions (Rodell et al., 2009; Famiglietti et al., 2011; Shamsudduha et al., 2012; Scanlon et al.,

2012; Taylor et al., 2013; Rodell et al., 2018). Given this unsustainable use of water, the question arises if future generations will still be able to benefit from the climate impact of irrigation.

In this study we assess if it is possible to sustain historical SM levels in the 21st century without over extracting local water resources. For this purpose, we develop a 'land water recycling' (LWR) scheme that irrigates the soil if SM level falls below late 20th-century conditions. The LWR scheme only uses local water sources, thus water is only applied to the soil if it is

available from runoff, and, potentially, a reservoir. We investigate if global-scale LWR is able to keep SM conditions at late 20th-century levels under future climate conditions and gauge its potential to mitigate local temperature extremes.



## 2 Methods

### 2.1 Model description

The Community Earth System Model (CESM, version 1.2, Hurrell et al., 2013) is a fully coupled Earth System Model, developed at the National Center for Atmospheric Research (NCAR). This state-of-the-art model has been extensively evaluated
(Hurrell et al., 2013; Meehl et al., 2013), and was used to study irrigation (Sacks et al., 2009; Hirsch et al., 2017; Thiery et al., 2017) as well as SM-climate feedbacks using SM prescription (Koster et al., 2004; Seneviratne et al., 2013; Hauser et al., 2017). We employ the Community Atmosphere Model, version 5.3 (Neale et al., 2012), and the Community Land Model, version 4.0 (CLM4.0, Oleson et al., 2010; Lawrence et al., 2011), with a horizontal resolution of $1.9° \times 2.5°$.

CLM4.0 is a $3^{rd}$-generation land surface model (Sellers et al., 1997; Pitman, 2003), solving the energy and water balance
of the land. The land surface is represented by five sub-grid land cover types (glacier, lake, wetland, urban and vegetated), where the vegetation is represented by up to 16 plant functional types. The soil is represented by 15 layers with exponentially increasing depth. Of these 15 layers only the first ten are hydrologically active, while the five deepest layers are only thermal slabs.

### 2.2 Land water recycling scheme

The aim of 'land water recycling' (LWR) is to keep SM conditions above a certain threshold, but only if water from local sources is available. Therefore, we develop a LWR scheme by extending an existing SM prescription module (Hauser et al., 2017), as illustrated in Figure 1. The LWR scheme adds water directly to each soil layer, analogously to drip-irrigation, using only water from runoff and, potentially, a reservoir. LWR is applied at every time step. First, the LWR scheme checks if SM is below the threshold. In this study, we use a historical SM climatology of the period 1971 to 2000 as threshold (Section 2.3).
It is calculated as the median SM value at each grid cell, soil level and day of the year. Next, the scheme checks if runoff is available and adds the required water to the soil, starting at the top most layer. If the water demand could not be satisfied from runoff, the scheme then uses water from the reservoir, if available. In turn, runoff that is left after applying water to the soil is used to fill up the reservoir, if it is not yet full.

### 2.3 Experimental design

We use CESM to generate four climate ensembles with three members, each. The first ensemble is a reference simulation (REF), forced with historical 'all-forcing' conditions from 1850 until 2005, and prolonged until 2099 with the Representative Concentration Pathway 8.5 scenario (RCP8.5; Meinshausen et al., 2011). Each ensemble member is branched off a long pre-industrial control simulation at a different year. This equates the standard CMIP5 setup of CESM. The reference simulations use an interactive ocean model (the Parallel Ocean Program model, version 2) to simulate ocean dynamics.



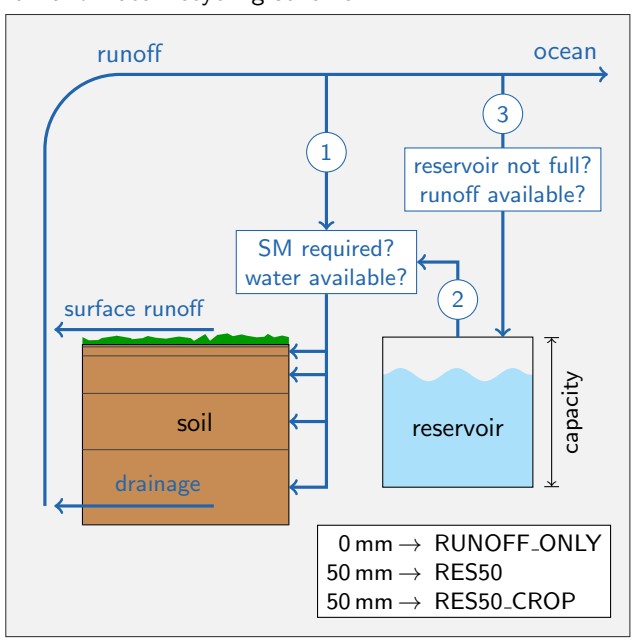

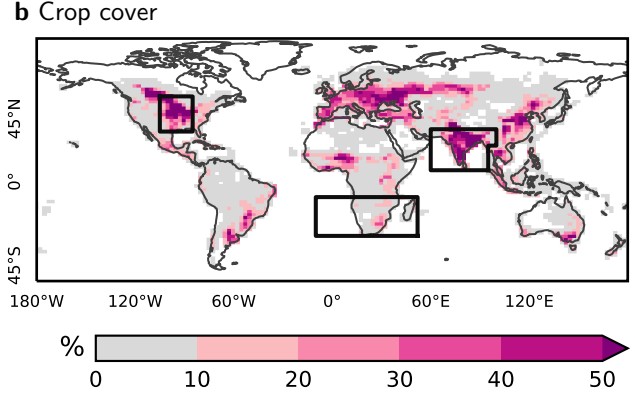

**Figure 1.** Land water recycling (LWR) scheme, list of experiments, and area where LWR is applied in the RES50_CROP experiment (see Section 2.3). (a) Illustration of the LWR scheme used in this study, and list of the reservoir capacity for each of the experiments. Numbers indicate the order of steps taken in the algorithm. (b) Crop fraction in CLM in the year 2000. In RES50_CROP, LWR is applied in all grid cells with more than 10 % crop fraction. The black boxes in (b) show three regions presented in Figure 5: Central North America (CNA), South Asia (SAS) and South Africa (SAF).





We conduct three further ensembles, RUNOFF_ONLY, RES50, and RES50_CROP, also summarized under the name 'experiments' (EXP). The experiments are branched off the reference simulations in 1950. In these simulations we apply the above-described LWR scheme. The SM target (Section 2.2) is the climatology of the first ensemble member of REF.

In the first sensitivity experiment, RUNOFF_ONLY, water is only taken from the runoff in the same grid cell, i.e. the reservoir capacity is $0\,\mathrm{mm}$ (Figure 1a). The second sensitivity experiment, RES50, includes a reservoir with a capacity of $50\,\mathrm{mm}$ at each grid cell. This allows to transfer a part of the water in time (e.g. from a wet spring into a dry summer), and should thus allow for a more reliable LWR. The size of the reservoir was chosen such that the resulting global reservoir capacity ($6284\,\mathrm{km}^3$) is close to the cumulative storage of human-built reservoirs as listed in the Global Reservoir and Dam (GRanD) database ($6197\,\mathrm{km}^3$) (Lehner et al., 2011). These first two sensitivity experiments are highly idealized in that they apply LWR globally to all vegetated and non-frozen land grid cells. They therefore gauge the potential of global-scale land water management. In a more realistic setting, the third sensitivity experiment, RES50_CROP10, is similar to RES50, but restricts LWR to all land areas with at least 10 % crop cover according to the PFT map of CLM4.0 (Figure 1b). In this experiment, LWR is thus mainly present in Europe, Central North America, and India. All sensitivity experiments prescribe sea surface temperatures (SSTs) and sea ice from the respective REF ensemble member to suppress impacts from changes in SSTs in response to LWR.

## 2.4 Analysis

All analyses are carried out on the pooled ensemble members, either using annual values or the mean over the respective regions' warm season, defined here as the three warmest consecutive months. We determine the warm season in REF for a historical period (1971 to 2000). The used warm "season" generally correspond to summer in mid- and high-latitude regions (Figure S1). But other time frames are found in some other regions, e.g. in the tropics.

In our analysis we focus on the end of the $21^{\mathrm{st}}$ century and calculate 30-year climatologies for 2070 to 2099. In light of the emerging literature on effects of limiting global warming to $1.5\,^\circ\mathrm{C}$ or $2.0\,^\circ\mathrm{C}$ above pre-industrial levels we also analyse how our experiments fare compared to half-a-degree additional warming in the global mean temperature. To this end we assess how much of the additional local warming is compensated by our sensitivity experiments. In particular, we calculate the relative cooling (or warming) of our experiments as:

$$\Delta T^* = \frac{T^*_{\mathrm{EXP},\,2.0} - T^*_{\mathrm{REF},\,2.0}}{T^*_{\mathrm{REF},\,1.5} - T^*_{\mathrm{REF},\,2.0}}, \tag{1}$$

where $T^*$ is a temperature index (see below), and 1.5 and 2.0 indicate the scenario with $1.5\,^\circ\mathrm{C}$, and $2.0\,^\circ\mathrm{C}$ global warming, respectively. We do not have dedicated experiments to determine the response at $1.5\,^\circ\mathrm{C}$ or $2.0\,^\circ\mathrm{C}$ warming. Therefore, we select years where REF experienced a mean global warming of $1.5\,^\circ\mathrm{C} \pm 0.15\,^\circ\mathrm{C}$, or $2.0\,^\circ\mathrm{C} \pm 0.15\,^\circ\mathrm{C}$ (Figure S2) with respect to 1861 to 1880. This selection criteria yields 19 years for the $1.5\,^\circ\mathrm{C}$ scenario and 18 years for the $2.0\,^\circ\mathrm{C}$ scenario.

To assess the influence of LWR on climate extremes, we compute three indices from the daily model output. They are: (i) the hottest daytime temperature of the year (TXx), measuring the intensity of heat extremes, (ii) the percentage of days that exceed the $90^{\mathrm{th}}$ temperature percentile (TX90p), i.e. the frequency of heat waves, and (ii) the duration of the longest heat wave per year when the 3-day running mean exceeds $90^{\mathrm{th}}$ temperature percentile (HWD), a measure of heat wave length. The $90^{\mathrm{th}}$



temperature percentile is calculated using the method of Zhang et al. (2005). We use a centred 15-day moving window for each calendar day of the year, pooling all three ensemble members. The 90[th] percentile is either calculated from the years 2070 to 2099, or 2023 to 2046 for the $1.5\,°C$ versus $2.0\,°C$ scenarios, such that the threshold and the exceedances are calculated for the same years.

Where appropriate we test for significance with a Wilcoxon-Mann-Whitney-U test (e.g. Wilks, 2011), as it is suited for non-Gaussian data distributions (e.g. for TXx). We conduct a significance test at each grid cell, which leads to an increased probability of falsely rejecting the null hypothesis (e.g. Wilks, 2016). This problem is overcome by applying the correction described in Benjamini and Hochberg (1995), using a global p-value of 5 %.

The influence of LWR on the surface temperature (TS) can be investigated with the help of the energy balance decomposition
(Luyssaert et al., 2014; Akkermans et al., 2014; Thiery et al., 2015; Hirsch et al., 2017). Taking the derivative of the surface energy balance yields the contribution of each term to the LWR-induced change in TS:

$$\Delta TS^4 = \frac{1}{4\epsilon\sigma TS^3}\left(\Delta SW_{net} + \Delta LW_{in} - \Delta LH - \Delta SH - \Delta R\right),\tag{2}$$

where $\epsilon$ is the surface emissivity, $\sigma$ the Stefan-Boltzmann constant, $SW_{net}$ the net short wave radiation, $LW_{in}$ the incoming (downward) long wave radiation, LH and SH are the latent and sensible head flux, and R is the residual term, which includes
the ground heat flux. $\Delta$ stands for the difference between the experiments and the reference simulation (EXP − REF). We examine the energy balance for regions defined in the Special Report on Managing the Risks of Extreme Events (SREX) (Seneviratne et al., 2012). We thereby focus on three regions: Central North America (CNA), South Asia (SAS) and South Africa (SAF) as shown in Figure 1b.

## 3    Results and discussion

### 3.1    Projected changes in soil moisture and hydrology

The LWR scheme only applies water to the soil if SM falls below the late 20[th]-century climatology. Therefore, it is of interest to assess the SM development in the 21[st] century. For the warm season of the year, CESM projects a strong decrease in surface SM, especially in Europe, North America, South Africa, and the north-east of South America, while most other regions show a small to moderate increase (Figure 2a). The response of CESM to climate change is consistent with the multi model median
projections from CMIP5 (Orlowsky and Seneviratne, 2013; Berg et al., 2017), with the exception of Australia. In Australia, the CMIP5 ensemble projects a SM decrease while CESM shows a small increase. Nonetheless, CESM can be viewed as a representative member of the CMIP5 ensemble.

LWR is able to substantially reduce the area with a negative SM trend (Figure 2b to d). In RUNOFF_ONLY, SM increases by 3 % (spatial median), whereas in REF soils dried out overall (−2.1 %). It is mainly the increase of SM in Europe and North
America which is responsible for this difference (Figure 2b). Allowing for storage of water for LWR further extends the area with a positive SM trend in the 21[st] century (Figure 2c). Finally, in RES50_CROP SM is almost only affected in areas where



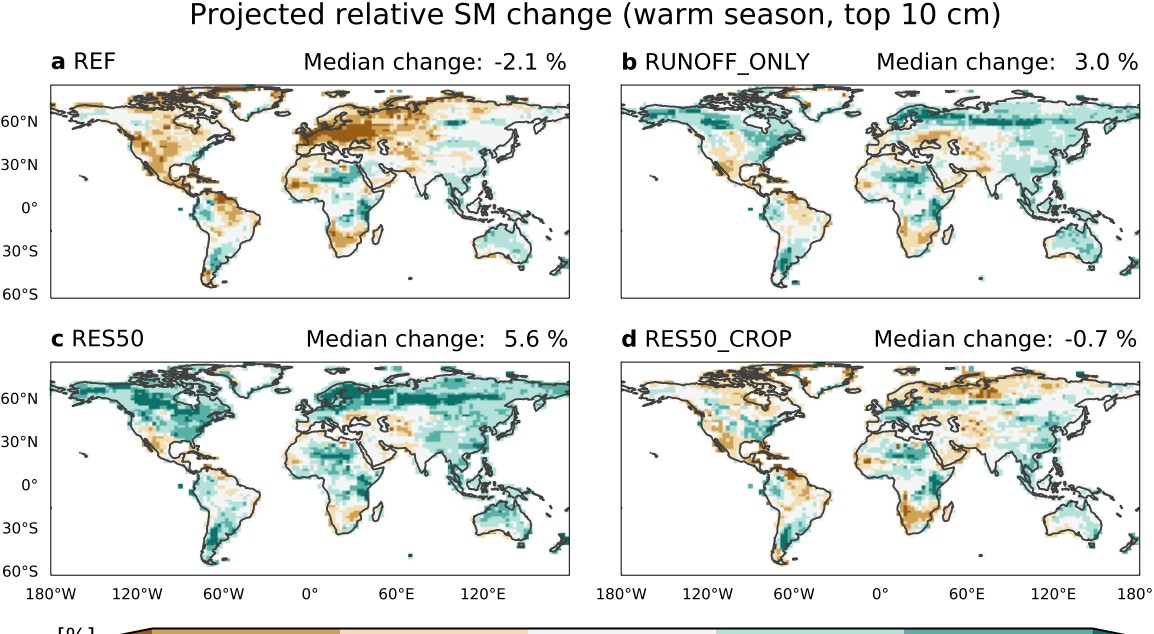

**Figure 2.** Projected change of SM in the topmost $10\,\mathrm{cm}$ of the soil relative to the soil moisture climatology (1971 to 2000) in REF. Only the warm season (three hottest consecutive months) are considered (Figure S1).

LWR is actually applied (not shown). This implies that there are no remote effects of LWR on SM. Overall, LWR is able to drastically reduce the fraction of days where the target SM conditions is not met (Figure S3).

LWR is not only expected to influence SM, but also other components of the hydrological cycle. In our analysis we will concentrate on precipitation, ET, runoff, and the introduced reservoir. Comparing precipitation at the end of the 21$^{\mathrm{st}}$ century

5 between the experiments and REF reveals some distinct patterns (Figure 3). Most areas in North America and Eurasia show a precipitation increase, while a large fraction of the tropics experiences a decrease. Partitioning this change into convective and large-scale precipitation shows that the former is responsible for the most of the signal (Figure S4). Averaged over all land areas precipitation is lower due to LWR (Table 1). This is compensated by increased precipitation over the oceans.

The regions with decreasing precipitation coincides to a large degree with monsoon regions (as defined in Zhang and Wang,

10 2008). A decrease in precipitation in monsoon regions due to irrigation was observed in earlier studies (Guimberteau et al., 2012; Puma and Cook, 2017; Thiery et al., 2017). It is consistent with a decrease in convective precipitation due to the cooling effect of the water management (Section 3.2). The precipitation increase in the extratropics, on the other hand, is likely due to the increased moisture input to the atmosphere, which can lead to intensification of the local hydrological cycle.

An increase in ET is expected when adding water to the soil, which is confirmed in Figure 3. This increase is present for

15 almost all land areas, and ranges between $1000\,\mathrm{km^3\,yr^{-1}}$ and $4000\,\mathrm{km^3\,yr^{-1}}$, depending on the experiment (Table 1). This



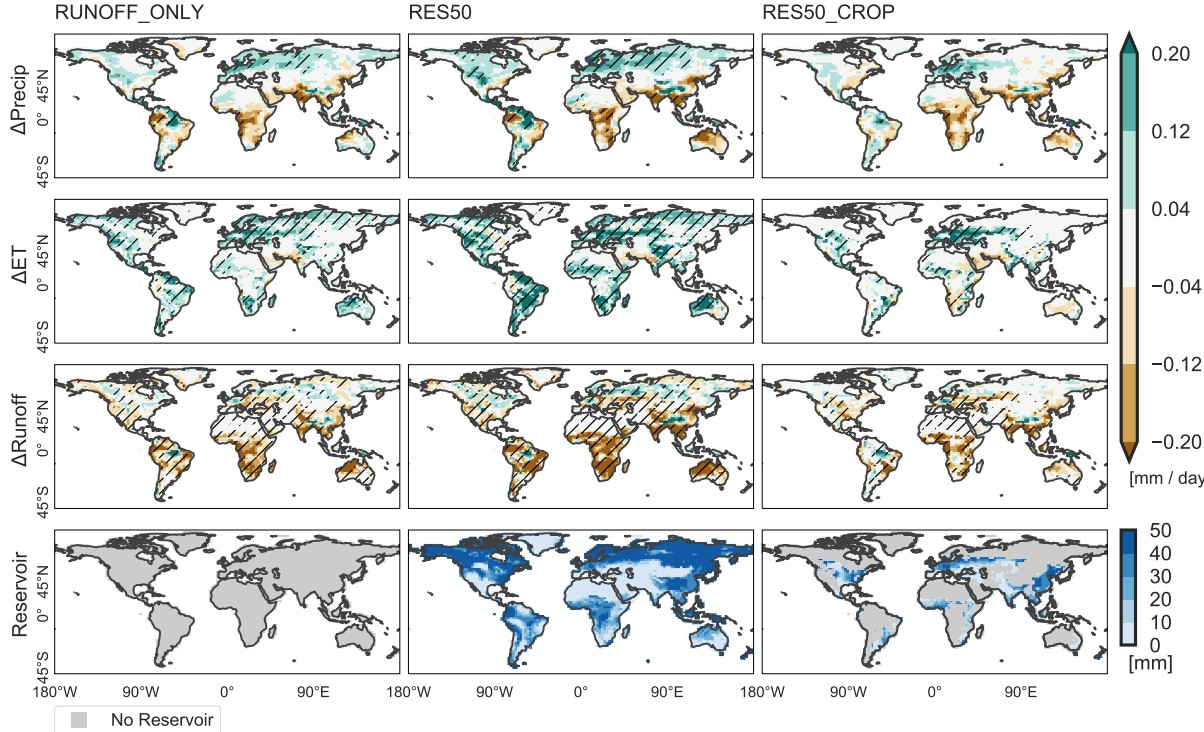

**Figure 3.** Difference between EXP and REF for precipitation (Precip), evapotranspiration (ET), and runoff. The fourth row shows the mean reservoir state. Hatching in the first three rows indicates grid cells with significant changes (Wilcoxon-Mann-Whitney-U test with a global p-value of 5 %). Note that runoff values for the experiments can become zero, which leads to significant runoff changes in some regions with very small absolute changes.

**Table 1.** Mean annual differences between EXP and REF for precipitation, ET, and runoff (in $\mathrm{km^3\,yr^{-1}}$).

| Variable | Domain | RUNFF_ONLY | RES50 | RES50_CROP |
|---|---|---|---|---|
| Precipitation | land | -1064 | -729 | -1696 |
| | ocean | 2178 | 3916 | 1533 |
| | global | 1114 | 3187 | -164 |
| ET | land | 2083 | 4032 | 1002 |
| | ocean | -968 | -881 | -1178 |
| | global | 1115 | 3150 | -176 |
| Runoff | land | -3157 | -4702 | -2680 |





is clearly higher than estimates from irrigation studies for RUNOFF_ONLY and RES50; (e.g. $418\,\mathrm{km^3\,yr^{-1}}$ in Thiery et al. (2017); $1233\,\mathrm{km^3\,yr^{-1}}$ in Sacks et al. (2009)). However, the LWR-induced ET surplus in RES50_CROP compares well with these previous model estimates.

Runoff is the only source of water for the LWR, for both direct water application and to fill the reservoir, thus we generally
expect a decrease. Indeed, globally runoff decreases by approximately two-thirds of the annual discharge of the Amazon river (e.g. Gupta, 2008). However, there are a some regions, mostly in the mid-latitudes, where the additional precipitation leads to a positive runoff signal (Figure 3).

The long-term (2070 to 2099) average of the reservoir implies that the reservoir is either (almost) full or empty (Figure 3). A spatial histogram also reveals a bi-modal distribution with one peak below $1\,\mathrm{mm}$ and the other at $47\,\mathrm{mm}$ for RES50 and $39\,\mathrm{mm}$
for RES50_CROP, respectively (not shown). However, most grid cells have a seasonal cycle of more than $10\,\mathrm{mm}$. Averaged over the whole SAS region, the reservoir contains $11\,\mathrm{mm}$ in the driest month while the water amount is twice as large in the wettest month $23\,\mathrm{mm}$. Similarly, the water in the reservoir fluctuates between $19\,\mathrm{mm}$ and $25\,\mathrm{mm}$ in CNA and between $7\,\mathrm{mm}$ and $20\,\mathrm{mm}$ in SAF. Thus, in many regions the reservoir is able to fulfil its function: providing water for LWR during the dry season of the year.

## 3.2   Land water recycling effect on temperature

Next we turn our attention to the temperature effect of LWR. The global annual land temperature is reduced by $-0.26\,\mathrm{^\circ C}$, $-0.42\,\mathrm{^\circ C}$ and $-0.23\,\mathrm{^\circ C}$ for RUNOFF_ONLY, RES50, and RES50_CROP (for 2070 to 2099), respectively. This is more than a recent estimate ($-0.05\,\mathrm{^\circ C}$) for realistic irrigation conditions (Thiery et al., 2017) during a historical period (1981 to 2010), but similar to a comparable SM prescription scheme ($-0.3\,\mathrm{^\circ C}$) (Hauser et al., 2017).

The effect of LWR on extreme temperature indices is shown in Figure 4. The intensity, $\Delta$TXx, is reduced by more than $-0.46\,\mathrm{^\circ C}$ over the global land area. Spatially, the reduction in TXx is more pronounced in the Northern- than in the Southern-Hemisphere, and Europe and Central North America stand out as hotspots of LWR-induced cooling. This stands in contrast to realistic irrigation experiments, where India experiences a strong cooling. This discrepancy can be explained by the soil moisture conditions in the historical period (1971 to 2000) and the projections (2070 to 2099). The soils in India are rather
dry in the historical simulations and there is no strong drying in the projections (Figure S5 and Figure S8). Therefore, our simulations do not apply much water in this region, leading to a small effect on temperature. In contrast, the dry soils are the reason for the high irrigation rates in this region, which results in the very strong cooling, but comes at the cost of strong groundwater depletion (Rodell et al., 2009).

For Europe and Central North America, on the other hand, REF indicates wet soils in the historical period and a strong
drying in summer in the next century (Figure S5). LWR is able to overcome this drying, especially when allowing for storage of water in a reservoir, causing the strong cooling in these regions. In historical irrigation simulations (as in Thiery et al., 2017) these regions do not receive as much water and therefore do not stand out as regions with a very strong cooling.

For the next two indices ($\Delta$TX90p and $\Delta$HWD) we restrict the analysis to the warm season of the year, as exceeding the $90^{\mathrm{th}}$ temperature threshold is not so relevant during the rest of the year. The change in the frequency of heat waves ($\Delta$TX90p)





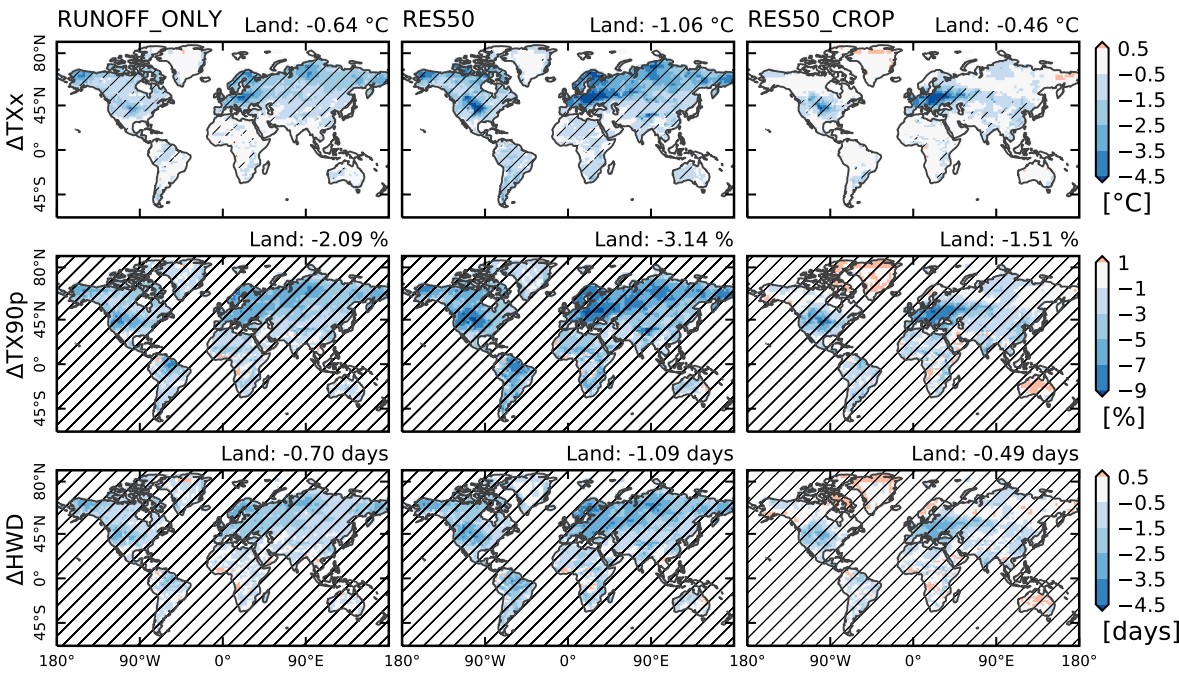

**Figure 4.** Difference between REF and RUNOFF_ONLY (left column), RES50 (central column), and RES50_CROP (right column) for temperature indices. Top row TXx (intensity), middle row TX90p (frequency) and bottom row HWD (duration). Hatching in the indicates significant grid cells (Wilcoxon-Mann-Whitney-U test with a global p-value of 5 %).

is between $1.51\,\%$ and $3.14\,\%$ when averaged over the whole land area (Figure 4, middle row). Given that $\Delta$TX90p is $10\,\%$ in REF (per definition), this is a substantial reduction. Indeed, in some regions the heat wave frequency is almost reduced to $0\,\%$ (not shown). Heat waves do not only become less frequent, but also shorter. The average length of a heat wave ($\Delta$HWD) is reduced by more than 0.5 days globally, and more than 3 days locally. For all three indices, it is evident that allowing for water

5 storage leads to a stronger cooling: RES50 is colder than RUNOFF_ONLY for virtually all land areas.

To better understand the background to the LWR-induced change in temperature, we make use of the energy balance decomposition introduced in Section 2.4. For CNA, there is a clear seasonal cycle in the surface temperature change (Figure 5). The strongest cooling effect of LWR occurs during the warm season and shortly after. The energy balance decomposition reveals that changes in the radiative budget are mostly responsible for the decrease in temperature. The decrease in $LW_{in}$ indicates a

10 higher cloud cover (Figure S6), in line with the observed increase in precipitation in this region (Figure 3). Land-atmosphere feedbacks only really kick in in July-to-September; that is, with one month offset compared to the considered warm season (three hottest months). These are the months with the driest soil in this region (not shown) – only then are the higher SM levels



able to increase the evaporative fraction and contribute to the cooling. Interestingly, the temperature response in RES50_CROP is smaller than RES50, even though LWR is applied at almost all grid cells in RES50_CROP in this region (Figure 1b). This may come from a smaller cloud cover increase in RES50_CROP (Figure S6), caused by the smaller water vapour input in other regions of the North American continent.

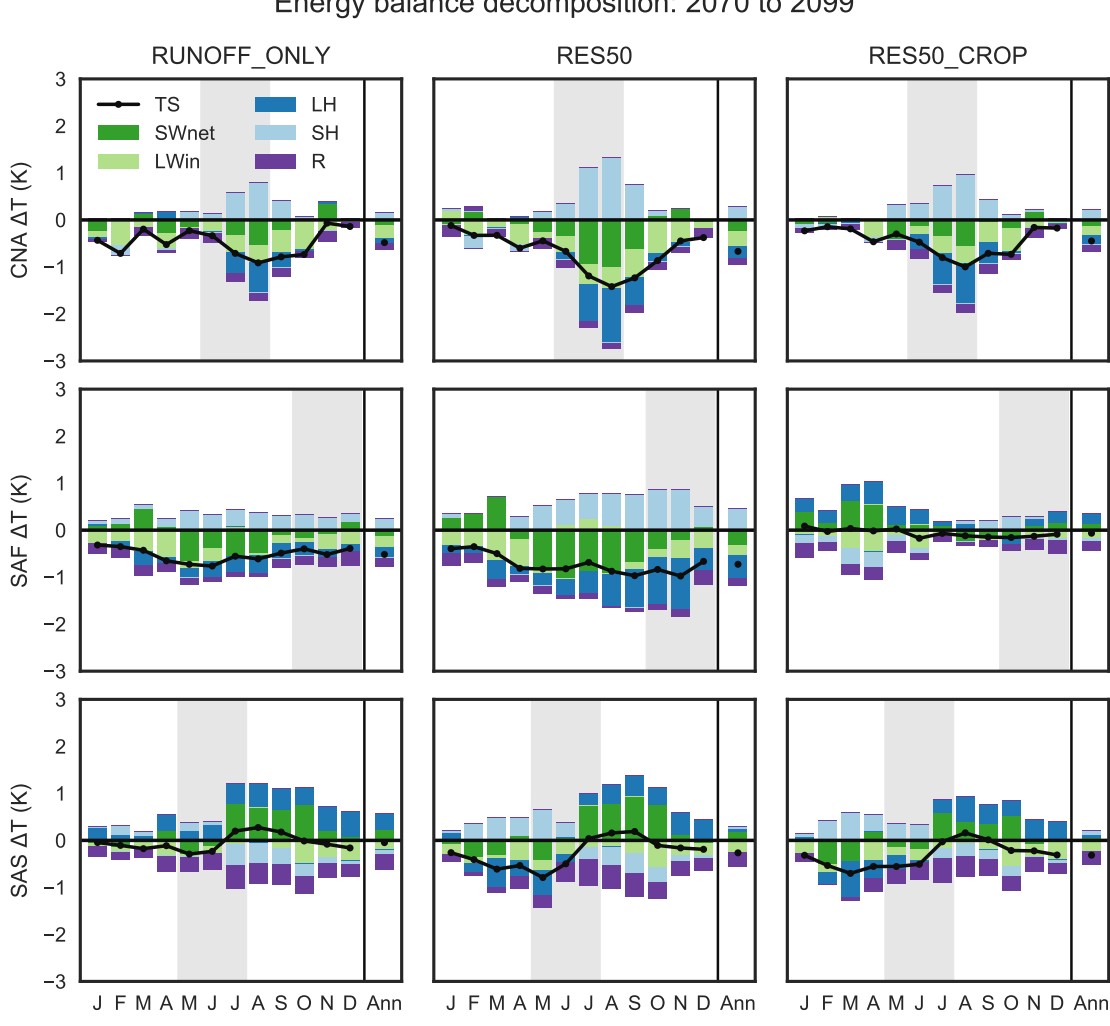

**Figure 5.** Seasonal cycle and annual mean (Ann) of surface temperature anomalies and energy balance decomposition for Central North America (CNA), South Asia (SAS) and South Africa (SAF). Grey shading indicates the three warm season.

5    SAF shows almost no seasonal cycle in TS and the contribution of the radiative- and land- terms of the energy balance seems to be more evenly distributed (Figure 5). In RES50_CROP this region has almost no grid cells where LWR is applied





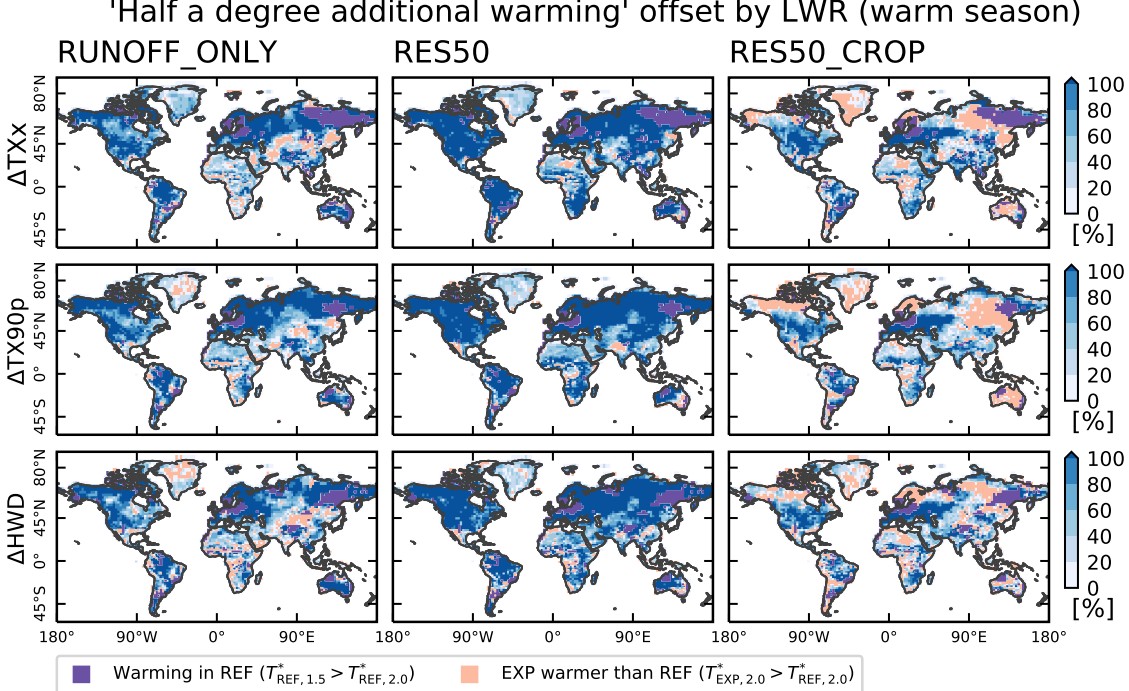

**Figure 6.** Offset of half-a-degree additional global mean warming by the LWR experiments, see Equation 2. Red patches indicate regions where the LWR experiments are warmer than REF (for 2.0 °C global mean warming). Magenta patches indicate regions where the 1.5 °C climate is warmer than the 2.0 °C climate in REF.

(Figure 1b), and consequently the temperature anomaly is small ($-0.1$ °C in the annual mean). Interestingly, the individual contributions are non-zero but compensate almost perfectly. For the next region, SAS, the response in surface temperature switches sign during the warm season. This change coincides with $SW_{net}$ reversing sign, consistent with the reduction of monsoon-rainfall and the associated decrease in cloud cover.

5    We could show that the developed LWR scheme reduces extreme temperatures – but is it also able to offset half a degree increase in the global mean temperature? We answer this question in Figure 6 with the help of Equation 2. The blue colors show the percentage warming offset due to LWR. Magenta patches indicate a warming in REF ($T^*_{REF, 1.5} > T^*_{REF, 2.0}$), and red patches show where EXP is warmer than REF ($T^*_{EXP, 2.0} > T^*_{REF, 2.0}$).

In RUNOFF_ONLY, LWR is able to offset the additional warming in parts of Eurasia, the Americas, and Australia, but not
10  in Africa and South Asia. Allowing for water storage (RES50) leads to a larger area where the LWR-induced cooling dominates over global mean warming. However, temperatures in Africa and the southern parts of Asia still remain warmer. Finally, in RES50_CROP the cooling effect is mostly restricted to the LWR-areas in Central North America and Europe.

Thus, our LWR scheme is able to locally offset the warming from half a degree additional warming. It does, however, not change the general warming trend due to raising greenhouse gases, which are almost the same in the LWR experiments and





REF (Figure S7). This is in accordance with Hirsch et al. (2017), and shows that increased irrigation may offer some respite from climate change, but does not alter the long term trends.

## 4    Conclusions

In this study we used idealized climate model experiments to study the potential effect of sustainable global-scale land water
management and its impact on temperature extremes. To this end, we developed a land water recycling (LWR) scheme that applies water to the soil if (i) it is dryer than in the 1971 to 2000 median soil moisture climatology and if (ii) water is available from local sources.

We compute four sets of climate model experiments with the Community Earth System Model with three ensemble members each. The four ensembles comprise a reference simulation and three sensitivity experiments including LWR. In the first sensi-
tivity experiment, LWR only applies water to the soil if runoff from the same time step is available. In the second sensitivity experiment water is also taken from runoff, but additionally a reservoir with a capacity of $50\,\mathrm{mm}$ is available such that e.g. surplus water that accumulates during the wet season can be used for LWR in summer. Finally, the third sensitivity experiment also uses runoff and a reservoir as water source, but only applies water in areas with a crop fraction of at least $10\,\%$.

We were able to show that LWR is able to maintain soil moisture conditions at late 20[th] century levels, for a large part of the
global land area. However, LWR also has a marked impact on the hydrological cycle. It leads to an increase in precipitation in mid- and high- latitudes, which is beneficial for areas where a precipitation decrease is projected for the next century. However, averaged over the global land area, this local increase is overcompensated by a reduction in precipitation in monsoon regions. As expected, LWR leads to a large-scale increase in evapotranspiration, due to higher soil moisture levels, and a decrease in runoff, as this is the only source of applied water. Further, the implemented reservoir is either 'full' ($>40\,\mathrm{mm}$) of 'empty'
($<5\,\mathrm{mm}$) in the long-term mean at most grid cells. Still, there is a strong seasonal cycle in the amount of water stored in the reservoir, indicating that it is able to provide water in the dry season.

LWR cools mean land air temperatures, but overall the effect is relatively small ($-0.23\,°\mathrm{C}$ to $-0.46\,°\mathrm{C}$). For mid-latitude regions the cooling results from a combination of increased cloud cover and an increase of the evaporative fraction. In monsoon regions the decrease in precipitation also goes along with a decrease in cloud cover, which increases the amount of incoming
solar radiation, offsetting part of the evaporative cooling. The impact of LWR on the upper end of the temperature distribution is larger. Annual maximum daytime temperatures, for example, decrease between $-0.46\,°\mathrm{C}$ and $1.06\,°\mathrm{C}$ over all land areas, and the frequency, and duration of heat waves is strongly reduced. For many regions LWR leads to a stronger cooling than half-a-degree additional global mean warming.

Adding a reservoir generally leads to more LWR and thus strengthens the response of the climate. This is especially well
visible in Central and South Europe, and Central North America. Precipitation projections for the 21[st] century indicate a strong decrease in these regions during the warm season of the year, but not for the whole year, rendering the reservoir especially effective. Restricting LWR to regions with at least $10\,\%$ crops, on the other hand, mostly restricts the influence to these regions.





While applying a water management scheme that affects the whole land area is certainly unrealistic, these sensitivity experiments can place an upper limit on the potential of LWR to mitigate climate change. Certain irrigation modules impose no limit on the water available for irrigation (e.g. Oleson et al., 2013). Thus, a potential avenue for future development is to couple a more realistic irrigation scheme with the water resource limitation presented in this study. The third experiment, restricting

LWR to crop areas, is a first step in this direction.

The LWR approach is in principle sustainable in the sense that it does not use more water than locally available from runoff, and as it does not lead to the depletion of groundwater reservoirs. Our scheme, however, imposes a large stress on runoff, leaving no residual flow in some regions. In practice this would have devastating ecological implications and imposing a minimum flow condition is a potential important addition to the LWR scheme. Further, while the total reservoir capacity in our

simulations compares well with observations (Lehner et al., 2011), the spatial distribution of this reservoir capacity is highly irregular. Accounting for this heterogeneity would require the development of a dedicated map with a per-grid cell reservoir capacity. A more realistic representation of reservoirs could also be achieved by including evaporation from the reservoir (Lowe et al., 2009; Dingman, 2015) or by adding an explicit reservoir operation scheme (Hanasaki et al., 2006).

Overall, we were able to show that sustainable land water management is theoretically able to keep soil moisture conditions

at late 20th levels. Our study opens a new perspective on how land water can influences local and regional climate. While LWR has only a small influence on mean temperatures, it leads to a substantial decrease in extreme temperatures, and can thus be seen as potential tool for local mitigation of climate change.

*Code availability.* The used code is available at https://github.com/IACETH/prescribeSM_cesm_1.2.x, where the documentation is linked. The code is released under a MIT licence. Revision 67cf64 was used to conduct the simulations with the land water recycling. Note that the

model framework (and code) of CESM/CLM is necessary to compile and use the code given in the repository.

*Competing interests.* The authors declare that they have no conflict of interest.

*Acknowledgements.* We thank and Urs Beyerle for support with CESM.





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
