# Peer review of "Potential of global land water recycling to mitigate local temperature extremes"

_Earth System Dynamics, 2018_

## Referee Comment (RC1) · Anonymous Referee #1 · 18 Jul 2018

General comments

In the present paper the authors assess the potential to keep soil moisture at a certain level by means of sustainable land water recycling (LWR), and analyze the impact on temperature extremes and on the hydrological cycle. A relatively simple (but conclusive) LWR scheme is introduced, and coupled to an Earth system model (CESM). Sensitivity experiments are carried out using different LWR settings. The results indicate that (in the present simulations) sustainable LWR (i) reduce the land area with decreasing soil moisture, (ii) lead to an increase of precipitation in mid-latitudes and a reduction in monsoon regions, and (iii) reduce hot temperature extremes.

I think that this is an interesting paper, which on the one hand analyses the impact of sustainable water management (irrigation), and on the other hand indicates how this

can be incorporated into Earth system models in a relatively easy way. The methodology is sound, the paper is well written and structured, and provides new and valuable results. Thus, I recommend publication. I have only minor comments the authors may like to consider.

Specific comments

1) P10L8-10: The authors state that changes in the radiation budget are responsible for the decrease in temperature, and a decrease of downward thermal radiation (LWin) indicates higher cloud cover. This seem to imply (perhaps unintentional) that the change in LWin is the most important factor. I may be wrong, but I would expect that higher cloud cover (and more moisture due to enhanced evapotranspiration) would increase the downward thermal radiation if the atmospheric temperature stays the same. Thus, the decrease in LWin may be a subsequent effect due to cooling of the atmosphere in response to a colder surface caused by higher evapotranspiration and less solar radiation (more clouds). This may need to be clarified.

2) P12L13-P13L2: In the sensitivity experiments SST and sea ice are prescribed. In my view this is a reasonable approach to analyze the (local) response for land areas, as it is done in most of the study. However, I think it is difficult to obtain robust conclusions for global and long-term properties (the global long term trend) without using interactive ocean and sea ice.

3) Precipitation: One main conclusion (and study-focus according to the title) is that LWR can reduce temperature extremes. However, also precipitation seems to change substantially. I'm wondering whether there is also a change of precipitation extremes. We may mitigate heat wave at the expense of having more flash floods in certain regions. Perhaps, the authors have looked at this, and may like to add a comment.

Technical corrections

1) P13L26: 1.06 -> -1.06 (?)

---

## Referee Comment (RC2) · Anonymous Referee #2 · 26 Nov 2018

Review of Hauser et al. esd-2018-48

I am recommending that this paper be accepted, subject to a few corrections and clarifications.

Overall I found the manuscript to be fairly straightforward. The authors analyzed their simulations very carefully and got a conclusion that, while having never really been demonstrated before, is perhaps unsurprising. I will say that it's difficult for me to get excited about this present paper.

What really interested me is lines 25-26 on page 2, as well as lines 27-29 on page 9. There is a fascinating paper to be written on how irrigation has suppressed climate change, and because groundwater is being depleted, accompanied by increasing de-

mand due to population growth, climate change is posed to accelerate in the near future. I realize that's a very different paper than what the authors wrote, and there isn't really too much wrong with the present paper, so I'm not going to suggest that they rewrite their entire paper to cater to my preference.

General comments:

I think some additional attention needs to be paid to caveats. The LWR scheme uses local water sources, but many of those water sources are already spoken for, generating competition among resources. This has important implications for agriculture, energy use, and transport. The authors are not well set up to address these implications (that's what integrated assessment models are for), but they can certainly discuss the importance of representing all of these processes and how they might affect the conclusions of the study.

Relatedly, the authors should discuss the feedbacks that their new scheme will have on the climate system. As an example, reducing runoff will reduce river flow, which will increase salinity in river deltas and reduce sediment transport. There are many other processes that I suspect are not included in this study. This needs to be mentioned.

Specific comments:

Page 1, line 21: "SM is prescribed to pre-defined values" such as? Page 2, line 4: What does "it" refer to? Page 2, line 10: Change "is" to "are" Page 2, line 16: "asymmetric" is misspelled Figure 1: I'm having trouble understanding panel a. The caption needs to be improved so I can better understand what is going on. Page 7, line 1: No strong remote effects. There are probably weak effects. Table 1: Experiment name is misspelled Page 9, line 6: "there are some regions" Page 9, line 23: I don't really understand this sentence. What realistic irrigation experiments? I thought your simulations were more realistic. Are you referring to anything in particular? In which case you need a citation.

---

## Author Comment (AC1) · 10 Dec 2018

**Response to reviewer #1 for "Potential of global land water recycling to mitigate local temperature extremes"**

We thank the reviewers for their positive comments and for the feedback, which helped us to improve the manuscript. In the revised version, we expanded the discussion on the limitations of our study and added a supplementary figure on extreme precipitation. Further, we made some minor improvements and corrections to the text, and added the land mean values to Figure 3.

**General comments**

In the present paper the authors assess the potential to keep soil moisture at a certain level by means of sustainable land water recycling (LWR), and analyze the impact on temperature extremes and on the hydrological cycle. A relatively simple (but conclusive) LWR scheme is introduced, and coupled to an Earth system model (CESM). Sensitivity experiments are carried out using different LWR settings. The results indicate that (in the present simulations) sustainable LWR (i) reduce the land area with decreasing soil moisture, (ii) lead to an increase of precipitation in mid-latitudes and a reduction in monsoon regions, and (iii) reduce hot temperature extremes.

I think that this is an interesting paper, which on the one hand analyses the impact of sustainable water management (irrigation), and on the other hand indicates how this can be incorporated into Earth system models in a relatively easy way. The methodology is sound, the paper is well written and structured, and provides new and valuable results. Thus, I recommend publication. I have only minor comments the authors may like to consider.

A1: We thank the reviewer for the encouraging comments.

**Specific comments**

1) P10L8-10: The authors state that changes in the radiation budget are responsible for the decrease in temperature, and a decrease of downward thermal radiation (LWin) indicates higher cloud cover. This seem to imply (perhaps unintentional) that the change in LWin is the most important factor. I may be wrong, but I would expect that higher cloud cover (and more moisture due to enhanced evapotranspiration) would increase the downward thermal radiation if the atmospheric temperature stays the same. Thus, the decrease in LWin may be a subsequent effect due to cooling of the atmosphere in response to a colder surface caused by higher evapotranspiration and less solar radiation (more clouds). This may need to be clarified.

A2: We agree with the reviewer - a higher cloud cover should go along with a higher LWin, given the same surface temperature. Thus, the lower LWin in our simulations is likely caused by the lower atmospheric temperature, while the smaller SWnet is due to the change in cloud cover. We will rewrite the paragraph as follows:

"The decrease in SWnet is caused by a higher cloud cover (Figure S7), in line with the observed increase in precipitation in this region (Figure 3). The lower LWin, on the other hand, is likely a response to the decreased boundary layer temperatures."

2) P12L13-P13L2: In the sensitivity experiments SST and sea ice are prescribed. In my view this is a reasonable approach to analyze the (local) response for land areas, as it is done in most of the study. However, I think it is difficult to obtain robust conclusions for global and long-term properties (the global long term trend) without using interactive ocean and sea ice.

A3: While we agree that there would be a feedback with the ocean, we argue that the temperature change is too small to substantially alter the long term trends. This is corroborated by the analysis of Hirsch et al., 2018, who showed that the trend in regional temperatures is similar with and without irrigation throughout the 21st century. We will update the paragraph as follows:

"Thus, our LWR scheme is able to locally offset the warming from half a degree additional warming. It does, however, not change the general warming trend due to rising greenhouse gases, which are almost the same in the LWR experiments and REF (Figure S8). This finding has to be taken with caution as we prescribe SSTs which will dictate the global mean warming. Nonetheless, they are in accordance with a similar study using an interactive ocean who also showed that the trend in regional temperatures is similar with and without irrigation throughout the 21st century (Hirsch et al., 2017)."

3) Precipitation: One main conclusion (and study-focus according to the title) is that LWR can reduce temperature extremes. However, also precipitation seems to change substantially. I'm wondering whether there is also a change of precipitation extremes. We may mitigate heat wave at the expense of having more flash floods in certain regions. Perhaps, the authors have looked at this, and may like to add a comment.

A4: We analysed annual maximum precipitation and will add Figure S5 as well a short discussion in the main text:

"We have limited the analysis of extreme precipitation to annual maximum 1-day precipitation amount (Rx1day, Figure S5). The detected changes in Rx1day between EXP and REF are generally smaller than 15 % and nonsignificant. The spatial pattern closely follow the change of mean precipitation shown in Figure 3."

**Technical corrections**

1) P13L26: 1.06 -> -1.06 (?)

A5: We will correct the mistake.

---

## Author Comment (AC2) · 10 Dec 2018

**Response to reviewer #2 for "Potential of global land water recycling to mitigate local temperature extremes**

We thank the reviewers for their positive comments and for the feedback, which helped us to improve the manuscript. In the revised version, we expanded the discussion on the limitations of our study and added a supplementary figure on extreme precipitation. Further, we made some minor improvements and corrections to the text, and added the land mean values to Figure 3.

I am recommending that this paper be accepted, subject to a few corrections and clarifications.
Overall I found the manuscript to be fairly straightforward. The authors analyzed their simulations very carefully and got a conclusion that, while having never really been demonstrated before, is perhaps unsurprising. I will say that it's difficult for me to get excited about this present paper.

What really interested me is lines 25-26 on page 2, as well as lines 27-29 on page 9. There is a fascinating paper to be written on how irrigation has suppressed climate change, and because groundwater is being depleted, accompanied by increasing demand due to population growth, climate change is posed to accelerate in the near future. I realize that's a very different paper than what the authors wrote, and there isn't really too much wrong with the present paper, so I'm not going to suggest that they rewrite their entire paper to cater to my preference.
B1: We thank the reviewer for the critical appraisal of the paper and the detailed comments.

**General comments**

I think some additional attention needs to be paid to caveats. The LWR scheme uses local water sources, but many of those water sources are already spoken for, generating competition among resources. This has important implications for agriculture, energy use, and transport. The authors are not well set up to address these implications (that's what integrated assessment models are for), but they can certainly discuss the importance of representing all of these processes and how they might affect the conclusions of the study.
B2: We agree with the reviewer that the LWR scheme uses water that would not be available in the real world. We already mention in our conclusions that our scheme could lead to a depletion of rivers and strong ecological impacts. We will expand on this, mentioning other competing interests:
"Our scheme, however, imposes a large stress on runoff, leaving no residual flow in some regions. In practice this would have devastating ecological implications and dramatically reduce river sediment transport (e.g. Chen et al., 2008). Additionally, some rivers are used for transport or to produce energy which would reduce the available water for LWR. Imposing a minimum flow condition is a potential important addition to the LWR scheme

(Jaegermeyr et al., 2017), which is expected to decrease the response of the climate system."

Relatedly, the authors should discuss the feedbacks that their new scheme will have on the climate system. As an example, reducing runoff will reduce river flow, which will increase salinity in river deltas and reduce sediment transport. There are many other processes that I suspect are not included in this study. This needs to be mentioned.

B3: We agree that many relevant hydrological processes, such as river temperature, quality and salinity, sediment transport, groundwater extraction and dam management, are not included in the current LWR scheme, nor are their potential feedbacks to the climate system. Regarding potential LWR impacts on ocean-atmosphere feedbacks, here we consciously choose to prescribe SSTs to minimize the need for a larger ensemble. We will add these considerations as a caveat to the conclusions:

Further, a number of potential earth system feedbacks arising from LWR are not considered in this study. For instance, LWR effects on hydrological processes, such as river temperature and salinity, water quality, sediment transport, groundwater extraction and dam management, are not included in the current LWR scheme and hence do not contribute to the overall climate feedbacks. In addition, we prescribe SSTs in our simulations, thereby disregarding potential feedbacks from the ocean. Performing simulations with an interactive ocean would for instance allow to assess the influence of changes in salinity due to the LWR, and compare the effects of less river water inflow to the ocean on the one hand, and enhanced precipitation and reduced evaporation over the ocean, on the other hand (Table 1).

**Specific comments**

Page 1, line 21: "SM is prescribed to pre-defined values" such as?

B4: There are a large number of different SM conditions that were used the literature (the plant wilting point, field capacity, simulated SM from a particular year, a climatological seasonal cycle, or a smoothed seasonal cycle). To investigate the influence of SM trends on temperature the most common is probably a climatological soil moisture. We will extend the sentence as follows:

The effect of future SM trends on temperature is typically assessed with idealised sensitivity experiments where SM is prescribed to predefined values, e.g a climatology (Koster et al., 2004; Seneviratne et al., 2013; Hauser et al., 2017).

Page 2, line 4: What does "it" refer to?

B5: We will rewrite the sentence to clarify the meaning:

A separate single-model experiment came to similar conclusions, identifying that in regions which experience drying, the SM feedback is responsible for up to one third of the projected increase in temperature extremes during the 21st century (Douville et al., 2016).

Page 2, line 10: Change "is" to "are"

B6: We will correct the mistake.

Page 2, line 16: "asymmetric" is misspelled

B7: We will correct the mistake.

Figure 1: I'm having trouble understanding panel a. The caption needs to be improved so I can better understand what is going on.
B8: Thanks for pointing this out. We will expand the caption and include a description of the algorithm in panel a as follows:
The blue lines indicate the 'flow' of water in the algorithm: surface runoff and subsurface drainage is combined to total runoff. If SM is below the target threshold, this total runoff is used to water the soil (1). In case there not enough runoff is available, water is taken from the reservoir (2). Finally, any remaining runoff is then used to fill up the reservoir if necessary (3). Note that steps (2) and (3) are only carried out if the reservoir capacity is >0 mm.

Page 7, line 1: No strong remote effects. There are probably weak effects.
B9: We agree with the reviewer and will change the sentence to:
This implies that there are no strong remote effects of LWR on SM.

Table 1: Experiment name is misspelled
B9: We will correct the mistake.

Page 9, line 6: "there are some regions"
B10: We will correct the mistake.

Page 9, line 23: I don't really understand this sentence. What realistic irrigation experiments? I thought your simulations were more realistic. Are you referring to anything in particular? In which case you need a citation.
B11: Thank you for pointing this out, we will change the sentence as follows:
This is in contrast to experiments with observed irrigation amounts, where India experiences a strong cooling (e.g. Thiery et al., 2017).